# The Effects of Sex and Strain on *Pneumocystis murina* Fungal Burdens in Mice

**DOI:** 10.3390/jof8101101

**Published:** 2022-10-19

**Authors:** Nikeya L. Macioce-Tisdale, Alan Ashbaugh, Keeley Hendrix, Margaret S. Collins, Aleksey Porollo, Melanie T. Cushion

**Affiliations:** 1Division of Infectious Diseases, Department of Internal Medicine, University of Cincinnati College of Medicine, Cincinnati, OH 45267, USA; 2Cincinnati VAMC, Medical Research Service, Cincinnati, OH 45220, USA; 3Center for Autoimmune Genomics and Etiology, Division of Biomedical Informatics, Cincinnati Children’s Hospital Medical Center, Cincinnati, OH 45229, USA; 4Department of Pediatrics, University of Cincinnati College of Medicine, Cincinnati, OH 45267, USA

**Keywords:** *Pneumocystis*, pneumonia, AIDS-related, sex differences, experimental design

## Abstract

Many preclinical studies of infectious diseases have neglected experimental designs that evaluate potential differences related to sex with a concomitant over-reliance on male model systems. Hence, the NIH implemented a monitoring system for sex inclusion in preclinical studies. Methods: Per this mandate, we examined the lung burdens of *Pneumocystis murina* infection in three mouse strains in both male and female animals at early, mid, and late time points. Results: Females in each strain had higher infection burdens compared to males at the later time points. Conclusion: Females should be included in experimental models studying *Pneumocystis* spp.

## 1. Introduction

The National Institute of Health Revitalization Act of 1993 mandated the inclusion of women participants in federally funded clinical research. With this inclusion of female participants, clinical research findings began to show differences in the drug response outcomes in women compared to men participants [1]. However, the use of females in experimental model systems was not widely adopted in preclinical research studies. A 2011 report noted the underrepresentation of female animals in 8 out of 10 biological disciplines [2]. Subsequently, in October 2014, the NIH implemented a monitoring system for sex inclusion in preclinical research, mandating federally funded researchers to report plans for the use of both male and female cell lines and animal models in preclinical studies or to explain why this was not necessary [3].

It is generally held that females develop more robust innate and adaptive immune responses than males and are less susceptible to infections caused by bacteria viruses, parasites, and fungi [4]. However, there are exceptions to this rule, and in some cases there have been contradictory results that were clearly dependent on the experimental model. For example, in a model of murine cutaneous leishmaniasis, males were more resistant to a physiological low-dose infection but more susceptible to higher doses and other methods of inoculation [5]. In another example, male and female mice differed in their responses to infection with *Plasmodium berghei*; whereas both sexes experienced peak parasitemia at the same time and time to death was the same, weight loss and brain responses were increased in males vs. females [6].

In the present report, we focused on sex differences associated with *Pneumocystis* pneumonia (PCP), an infection caused by pathogenic yeast-like fungi in the genus *Pneumocystis*. In humans, PCP infects males and females with compromised immune systems and can cause life-threatening pneumonia. Strikingly, there was but a single publication in the PubMed database that compared PCP outcomes in men and women with HIV in 1987 showing women were more likely than men to die in the hospital from PCP [7]. A more recent meta-analysis showed an increase in PCP in non- immunodeficiency virus (HIV)-infected patients, where being a female non-HIV patient was a risk factor, resulting in an increased mortality rate in female vs. male non-HIV patients [8].

Investigators have generally used male rodents for studies of *Pneumocystis* infection, especially for preclinical drug development [9]. The identification of new prophylactic and therapeutic anti-*Pneumocystis* agents is a critical focus of investigation, as there are few drugs with which to treat PCP. It is not known whether there are differences in responses by the sexes to this infection, but it could be an important consideration in future experimental designs. In this report, we compared the fungal burdens of *P. murina* in dexamethasone-immunosuppressed male and female mice to address a fundamental aspect of the mouse model of infection.

In the present study, we assessed fungal burdens by total nuclei counts, which include all life cycle stages of *Pneumocystis* species, and by enumeration of asci. Asci are fewer in number in most *Pneumocystis* infections, and are thought to be the agent of transmission.

## 2. Material and Methods

### 2.1. Design

Corticosteroid-immunosuppressed male and female Balb/c, C3H/HeN, and C57BL/6 mice (Charles River Laboratories) were infected with *P. murina* through exposure to *P. murina*-infected and immunosuppressed mice (seed mice) of the same strain and gender, following previous protocols [10,11]. Recipient mice were 6 weeks old at the time of infection. Seed mice have existing fulminate infections and transmit the infection to the naïve immunosuppressed mice through an airborne method of transmission [10]. The average fungal burden of the seed mice was 5 × 10^7^ nuclei/lung. One seed mouse can infect up to four experimental mice at a time. To control for varying *P. murina* burdens in the seed mice, they are rotated throughout the experimental mouse cages 4 times over a 2-week period. The recipient (and seed mice) mice were immunosuppressed by adding dexamethasone at 4 µg/liter to their drinking water and housed with the seed mice for 2 weeks. The mice were fed autoclaved standard lab chow from Charles River Laboratories (Wilmington, MA, USA) and administered acidified water (1 MHCL) to discourage secondary microbial infections, ad libitum. At 4, 6, and 8 weeks post-exposure, 8 male and female mice of each strain were sacrificed. Asci and total nuclei were enumerated as describe [12]. Microscopic enumeration of asci assesses the numbers of this life cycle stage, which is alleged to be the product of the sexual cycle of these fungi as well as its transmissive stage. Enumeration of nuclei represents total fungal burden, as the nuclei of all life cycle stages are counted.

### 2.2. Evaluation of Organism Burden

At each of the time points, mice were euthanized by carbon dioxide inhalation per IACUC approved methods, and the lungs were removed. After dissection from the bronchi, lungs were homogenized using the gentleMACS dissociator from Miltenyi Biotec (Auburn, CA, USA), diluted in phosphate buffered saline, centrifuged at 250 rpm for 10 min, treated with aqueous ammonium chloride to remove host cells, and used to prepare slides for enumeration [11].

### 2.3. Microscopic Enumeration

Slides were made by dropping 10 μL of the homogenized lungs onto glass microscope slides within a circumscribed area and allowed to air dry. The slides were heat fixed and stained with Cresyl Echt Violet (CEV) (Thermo Fisher Scientific, Waltham, MA, USA) for enumeration of asci and rapid Wright-Giemsa (Thermo Fisher Scientific) for enumeration of all life cycle stages. *P. murina* asci and nuclei were enumerated by counting 30 microscopic fields at 1250× power [13]. Data were expressed as log_10_ mean ± standard deviation per lung. GraphPad Prism v. 5 [12] was used to determine significance using the unpaired *t*-test.

### 2.4. Survival Curve

At each time point or depending on the health of the mouse, 8 mice were sacrificed per group. Mice that exhibited signs of labored breathing, reduced response to external stimuli, or reduced ambulation were humanely euthanized due to declining health. The survival curve represents any deaths that occurred prior to the 8-week endpoint. The survival of each group was analyzed by GraphPad Prism v.5 using the log-rank (Mantel–Cox) test and the Gehan–Breslow–Wilcoxon test. We typically stop these experiments at or around 8 weeks due to the poor condition of the mice and humanely sacrifice them at this endpoint.

## 3. Results and Discussion

Male and females in three mouse strains were evaluated for *Pneumocystis* burden at three time points that represent early, mid, and late infection (Figure 1 and Figure 2). In Figure 1, the data were parsed to compare fungal burdens in male and female groups within each strain. Significant differences were observed in total lung burden (Figure 1A, “nuclei counts”) at 6 weeks for all gender pairs, with females carrying larger burdens. After 8 weeks of immunosuppression, the total organism burdens were significantly higher in the C3H/HeN and the C57BL/6 female mice. No significant differences were observed in the total lung burdens at the early (4-week) time point. Interestingly, only a single significant gender difference was observed in asci numbers between male and female mice and that was at 6 weeks in the C3H/HeN strain (Figure 1B, “Asci Counts”). Since the nuclei counts represent all life cycle stages, including the asci, these results may indicate an increased trophic burden in females vs. males in the other two strains.

Next, the *P. murina* burdens were compared within the same gender in the different mouse strains (Figure 2). In terms of total lung burden, the C57BL/6 female mice had significantly lower burdens at all three time points compared to females in the other two strains (Figure 2A). The Balb/c and C3H/HeN female mice had a significant difference only at 6 weeks (Figure 2A). Few differences were notable for numbers of asci in the female mice across strains (Figure 2B), apart from lower numbers in the C57BL/6 mice at 4 and 6 weeks, and with the BAlb/c females having had higher numbers of asci at 8 weeks.

Analyzing the total lung burdens in male mice across strains, males of the C57BL/6 strain had burdens significantly lower at the 4- and 8-week time points, like their female counterparts (Figure 2C). Although burdens at 6 weeks in this group were lower than those of males in the other strains, significance was not reached (Figure 2C). The total fungal burdens in Balb/c male mice were significantly higher than males of the other two strains at the 6- and 8-week time point (Figure 2C). In the male mice, asci burdens were statistically different in the BAlb/c mice at 6 and 8 weeks, where again, there were higher numbers than in the other two strains (Figure 2D).

The males and females in the C3H/HeN strain experienced a higher mortality rate than the other strains (46% vs. 100%, respectively) (Figure 2E), with all female mice succumbing to the infection or having had to be sacrificed due to morbidity prior to the end of the experiment. Note that, in this animal model, all mice are sacrificed at the terminal endpoint of 60 days due to morbidity associated with the infection. Survival curves are based on mice who expired prior to the endpoint.

Potential limitations of the study included the co-housing (“seed”) model of infection. Though this airborne transmission model replicates the actual mode of infection, it is possible that there were slight differences in the fungal burdens of the transmitting mice. However, we showed previously that the transmission from infected rodents to uninfected rodents required less than 10 organisms to elicit the pneumonia [14]. In addition, the same infected seed mice could not be used to infect both sexes, as combining sexes could lead to death or injury. Since we did not weigh the mice throughout the infection, there may have been sex-related differences in response to the corticosteroids that could have influenced the observed differences.

Adhering to the NIH sex inclusion policy, we showed that gender did influence the progression of *P. murina* infection in this study, with higher fungal burdens observed in females of all three mouse strains at the mid-time (6 weeks) point and for C3H/HeN and C57/BL6 female mice after 8 weeks of immunosuppression. Reasons for this gender disparity could relate to factors in the lung environment, differential responses to the immunosuppressive regimen, or in the immune response to these fungal pathogens. It is interesting to note that few differences were observed in the numbers of asci regarding strain or gender. Balb/c male or female mice were the most permissive to this fungal infection, as evidenced by the highest lung burdens. The outcomes from this study should guide future experimental design in pathogenesis of infectious diseases; systems biology “omics” approaches; and computational models of disease progression. The bases of these gender differences should be explored in future studies. Notably, these findings reiterate the importance of examining the effects of disease progression in both sexes, since such differences could impact clinical response to anti-*Pneumocystis* therapy or indicate distinct pathologies.

## Figures and Tables

**Figure 1 jof-08-01101-f001:**
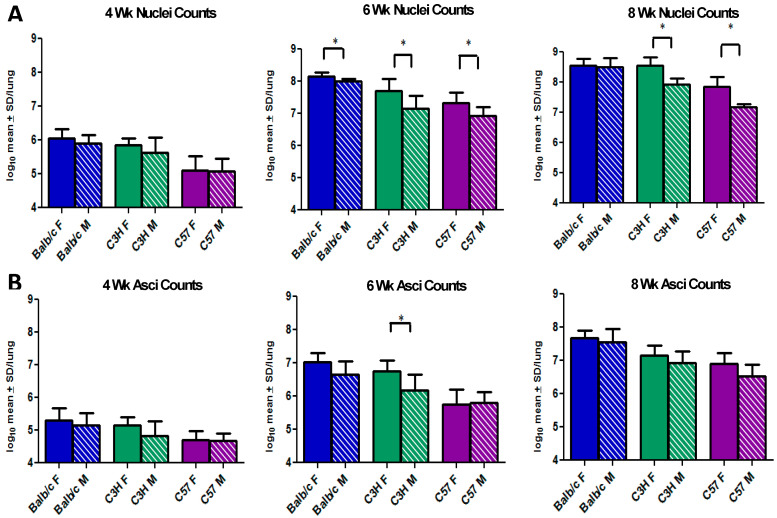
***Pneumocystis murina* burdens comparing males and females in 3 mouse strains.** At 4, 6, and 8 weeks, asci and nuclei (total fungal burden) of male (M) and female (F) Balb/c, C3H/HeN (C3H), and C57BL/6 (C57) were quantified by microscopic enumeration, log transformed, and expressed as the log_10_ mean ± the standard deviation per lung (Y-axis). (*) Indicates statistical significance at *p* < 0.05 using an unpaired *t*-test between male and female mice of the same strain. Panel **A**: Female vs Male nuclei counts in the 3 mouse strains over the 3 time points; Panel **B**: Female vs Male asci counts in the 3 mouse strains over the 3 time points.

**Figure 2 jof-08-01101-f002:**
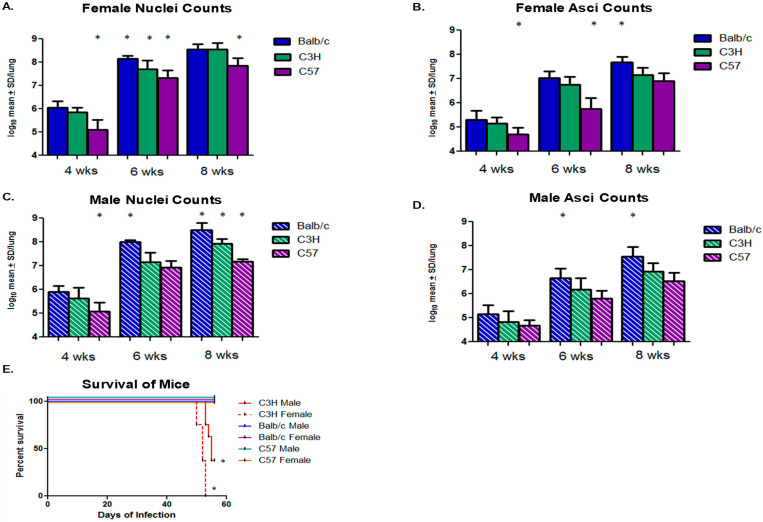
***Pneumocystis murina* burdens comparing the female and male sexes of each mouse strain at the three time points and survival curve of the study**. The same data presented in Figure 1 were parsed to examine differences among the sexes in each mouse strain. (**A**) Female nuclei count. * Indicates statistical significance of C57 mice nuclei burdens at 4 and 8 weeks compared to other strains. (**B**) Female asci count. * Indicates statistical significance of asci burdens of female C57 mice compared to the other strains at these time points. (**C**) Male nuclei count. * Indicates statistically significant differences in organism burdens. (**D**) Male asci count. * Indicates statistically significant asci burdens of Balb/c mice when compared to other strains at 6 and 8 weeks. All statistical significance was set at *p* < 0.05 using one-way ANOVA and Newman–Keuls multiple comparison test. (**E**) Survival curves for all mouse strains prior to the terminal time point. Survival curves were analyzed using GraphPad Prism v.5. (*) Indicates statistical significance at *p* < 0.0088 for the male C3H/HeN and *p* < 0.0001 for the female C3H/HeN.

## Data Availability

The data that support the findings of this study are available from the corresponding author, [M.T.C.], upon reasonable request.

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
