# Peer review of "The Effects of Sex and Strain on Pneumocystis murina Fungal Burdens in Mice"

_jof, 2022, doi:10.3390/jof8101101_

Round 1
Reviewer 1 Report
Overall this manuscript is well written and provides an essential body of knowledge that can inform the field and have an impact on future experimental design. The manuscript can be improved with a few minor corrections and warrants publication.
Lines 62-67: What are the age ranges of the mice at the time of infection? Some details on the seed mice would also be helpful for those unfamiliar with this method of infection, i.e. how long had seed mice been infected at the time of co-housing and what is their typical pneumocystis burden?
Lines 77-83: (Suggest more detail/reference): Whole lungs were homogenised for microscopy preparation. In previous experience, are the weights of lungs in the mice being compared (male vs female, between strains) the same, i.e. no statistically significant differences? Could a difference in lung weights not lead to different concentrations of pneumocystis in 10 microlitre slide preparations?
Line 92-97: Is the weight loss of mice recorded during the course of the experiments? Please provide details/reference on how the “health of the mouse” (line 93) is evaluated. This section is confusing, why would survival mice be killed at specific time-points, please clarify.
Line 131: (46 vs. 100%) add “respectively”
Lines 136-139: (suggestion for more discussion): There was a difference in pneumocystis burden at 6 weeks, but not at 8 weeks, between male and female BALB/c mice. Do the authors think it is possible for a similar reversal to happen in the other strains, if the experiment had been extended to 10-12 weeks? This might reflect a difference in pneumocystis growth dynamics in the different strains. Perhaps the authors could comment on this in the discussion.
Furthermore, the authors could add to the discussion a potential limitation with the seed model when using both sexes. Males and females cannot be combined when exposed to seed mice of the same sex. This may have influenced the timing of infection and potentially influence the results. This can be mentioned in the discussion to clarify.
Figure 1 and 2; consider correcting C57 to C57BL6 (as in the text) for clarity.
Figure 2 legend: “The data presented in Figure 1 were parsed for gender and strain differences”. Please clarify the reference to Figure 1 in this legend.
Pneumocystis murina should be in italics throughout.
Author Response
Overall this manuscript is well written and provides an essential body of knowledge that can inform the field and have an impact on future experimental design. The manuscript can be improved with a few minor corrections and warrants publication.
Lines 62-67: What are the age ranges of the mice at the time of infection? Some details on the seed mice would also be helpful for those unfamiliar with this method of infection, i.e. how long had seed mice been infected at the time of co-housing and what is their typical pneumocystis burden?
The mice at time of infection were 6 weeks old. The seed mice used were infected with P. murina for 6 weeks and their average infection burden was 5 x 10^7 nuclei/lung.
This has been noted in Lines 65-68.
Lines 77-83: (Suggest more detail/reference): Whole lungs were homogenised for microscopy preparation. In previous experience, are the weights of lungs in the mice being compared (male vs female, between strains) the same, i.e. no statistically significant differences? Could a difference in lung weights not lead to different concentrations of pneumocystis in 10 microlitre slide preparations?
We don’t weigh the lungs as a consistent level of infection has resulted from this process in our hands, as reported in the earlier literature (Powles et al. Infect. Immun. Apr. 1992, 1397-1400;
Line 92-97: Is the weight loss of mice recorded during the course of the experiments? Please provide details/reference on how the “health of the mouse” (line 93) is evaluated. This section is confusing, why would survival mice be killed at specific time-points, please clarify.
We don’t weigh the mice. Mice that exhibit signs of labored breathing, reduced response to external stimuli and reduced ambulation are humanely euthanized due to declining health. The survival curve represents any deaths that occurred prior to the 8 week endpoint. This is stated in the results/discussion, lines 134-136.
We have also added the criteria to determine poor health, lines 95-97
Line 131: (46 vs. 100%) add “respectively”
Added, line 135
Lines 136-139: (suggestion for more discussion): There was a difference in pneumocystis burden at 6 weeks, but not at 8 weeks, between male and female BALB/c mice. Do the authors think it is possible for a similar reversal to happen in the other strains, if the experiment had been extended to 10-12 weeks? This might reflect a difference in pneumocystis growth dynamics in the different strains. Perhaps the authors could comment on this in the discussion.
It is an interesting question, however, we cannot extend this experiment out to 10-12 weeks due to the progression of infection and associated morbidity and mortality of the mice.
Furthermore, the authors could add to the discussion a potential limitation with the seed model when using both sexes. Males and females cannot be combined when exposed to seed mice of the same sex. This may have influenced the timing of infection and potentially influence the results. This can be mentioned in the discussion to clarify.
This is a good point and we have included a paragraph regarding potential limitations of the study: lines 140-146.
Figure 1 and 2; consider correcting C57 to C57BL6 (as in the text) for clarity.
Explanation of each abbreviation is included in the legend.
Figure 2 legend: “The data presented in Figure 1 were parsed for gender and strain differences”. Please clarify the reference to Figure 1 in this legend.
The legend of Figure 2 has been re-written for clarification.
Pneumocystis murina should be in italics throughout.
Done.
Reviewer 2 Report
Steroid-treated male and female mice are both highly susceptible to P. murina infection, however the data supports the conclusion that female mice of the strains used in the study have higher fungal burdens than male mice.
1. Were the total burdens for each mouse normalized to mouse lung weight? Since there are likely significant size differences between the sexes, the investigators might be underestimating the differences in burden.
2. Did the investigators observe any changes in the rate of weight loss over the course of the study? This might indicate if there were gender-associated differences in the responsiveness to corticosteroids that might affect the level of infection. Variable responsiveness to the corticosteroids could factor into the observed differences.
3. How did the investigators control for the level of infection of the seed mice? Were male experimental mice housed with male seed mice, and female with female? These details should be included.
Author Response
Steroid-treated male and female mice are both highly susceptible to P. murina infection, however the data supports the conclusion that female mice of the strains used in the study have higher fungal burdens than male mice.
- Were the total burdens for each mouse normalized to mouse lung weight? Since there are likely significant size differences between the sexes, the investigators might be underestimating the differences in burden.
Good point but we did not weigh the mice.
- Did the investigators observe any changes in the rate of weight loss over the course of the study? This might indicate if there were gender-associated differences in the responsiveness to corticosteroids that might affect the level of infection. Variable responsiveness to the corticosteroids could factor into the observed differences.
Unfortunately, the mice were not weighed throughout the study. This point has been added to the potential limitations paragraph, lines 148-150.
- How did the investigators control for the level of infection of the seed mice? Were male experimental mice housed with male seed mice, and female with female? These details should be included.
Males experimental mice were only seeded with male mice and vice versa. Seed mice were sacrificed after seeding was done and they had an average P. murina burden of 5 x 10^7 nuclei/lung. This has been included in this revised manuscript, lines 65-68.
Reviewer 3 Report
This study aims to evaluate the impact of mice strain and mice gender on the outcome of Pneumocystis infection. Globally, they describe higher fungal burden in female mice compared to male and higher fungal burden in Balb mice compared to C57bl6 mice.
This is an interesting results as this has not been previously clearly published, although mice model is the main animal model to study Pneumocystis.
I have one main concern and few comments:
-There is no details on the seed mice : number of seed mice for the number of evaluated mice? There is no control of the fungal burden of these seed mice? Although the authors have chosen corresponding gender and strain for the seed mice, there could be individual cases of high burden in the seed mice. They could have a major impact on the results if there is 1 seed mouse for 4 evaluated mice for instance.
-l 109-110 : "these results may indicate an increased 109 trophic burden in females vs. males in the other 2 strains". To strengthen this remark, could the authors add the ratio of asci/nuclei for each mouse?
- do the authors have clinical evaluations of the mice? other than mortality? Weight, mobility, tachypnea, mouse fur?
- The discussion is very small and there are no references. Notably on the well-known difference of inflammation response depending on the strain, with higher Th1 vs Th2 in C57bl6. And there are previous studies showing that Th1 could be useful to improve Pneumocystis elimination. For gender/inflammation?
- Also there are few information on the two stages of Pneumocystis (asci and trophozoites) either in the introduction or discussion. However, it is one of the main results.
-line 130 : did you mean survival rate? (100%)
- Pneumocystis should be italicized throughout the text.
Author Response
This study aims to evaluate the impact of mice strain and mice gender on the outcome of Pneumocystis infection. Globally, they describe higher fungal burden in female mice compared to male and higher fungal burden in Balb mice compared to C57bl6 mice.
This is an interesting results as this has not been previously clearly published, although mice model is the main animal model to study Pneumocystis.
I have one main concern and few comments:
-There is no details on the seed mice : number of seed mice for the number of evaluated mice? There is no control of the fungal burden of these seed mice? Although the authors have chosen corresponding gender and strain for the seed mice, there could be individual cases of high burden in the seed mice. They could have a major impact on the results if there is 1 seed mouse for 4 evaluated mice for instance.
1 seed mouse can infect up to 4 experimental mice at a time. To control for varying P. murina burdens in the seed mice, they are rotated throughout the experimental mouse cages 4 times over a 2-week period.
These statements have been added, lines 68-71
-l 109-110 : "these results may indicate an increased 109 trophic burden in females vs. males in the other 2 strains". To strengthen this remark, could the authors add the ratio of asci/nuclei for each mouse?
This is a good point, but we no longer have the slides to perform these counts.
- do the authors have clinical evaluations of the mice? other than mortality? Weight, mobility, tachypnea, mouse fur?
Mice that exhibit signs of labored breathing, reduced response to external stimuli and reduced ambulation are humanely euthanized due to declining health. These criteria have been added to the text, lines 98-100.
- The discussion is very small and there are no references. Notably on the well-known difference of inflammation response depending on the strain, with higher Th1 vs Th2 in C57bl6. And there are previous studies showing that Th1 could be useful to improve Pneumocystis elimination. For gender/inflammation?
We are puzzled by the statement regarding the lack of references. There were 13 and we have added another to make 14 references. Inflammation in each sex was not the focus of these studies, but it would be an interesting follow-up study.
- Also there are few information on the two stages of Pneumocystis (asci and trophozoites) either in the introduction or discussion. However, it is one of the main results.
Good point. An explanation is included in lines 60-63.
-line 130 : did you mean survival rate? (100%)
No. More died.
- Pneumocystis should be italicized throughout the text.
It has been, though this italicization varies among journals.